

# Sensitivity of Aeolus HLOS winds to temperature and pressure specification in the L2B processor

Matic Šavli[1], Vivien Pourret[1], Christophe Payan[1], and Jean-François Mahfouf[1]

[1]CNRM, Université de Toulouse, Météo-France, CNRS, Toulouse, France

**Correspondence:** Vivien Pourret (vivien.pourret@meteo.fr)

**Abstract.** The retrieval of wind from the first Doppler wind lidar of Europen Space Agency (ESA) launched in space in August 2018 is based on a series of corrections necessary to provide observations of a quality useful for Numerical Weather Prediction (NWP). In this paper we examine properties of the Rayleigh-Brillouin correction necessary for the retrieval of horizontal line-of-sight wind (HLOS) from a Fabry-Perot interferometer. This correction is taking into account the atmospheric stratification,

namely temperature and pressure information that are provided by a NWP model as suggested prior launch. Since NWP models contain errors the main goal in the study is to evaluate the impact of these errors on the HLOS sensitivity by comparing the Integrated Forecast System (IFS) and Action de Recherche Petite Echelle Grande Echelle (ARPEGE) global model temperature and pressure short term forecasts collocated with the Aeolus orbit. The model error is currently not taken into account in the computation of the HLOS error estimate since its contribution is believed small. This study largely confirms this statement to

be a valid assumption, although it also shows that model errors could locally (i.e. jet-stream regions, below 700 $\mathrm{hPa}$ over both earth poles and as well in stratosphere) be significant. For a future Aeolus follow-on missions this study suggests to consider realistic estimations of model errors in the HLOS retrieval algorithms, since this will lead to an improved estimation of the Rayleigh-Brillouin sensitivity uncertainty contributing to the HLOS error estimate and better exploitation of space lidar winds in NWP systems.

## 1  Introduction

The ESA's Aeolus wind satellite has been launched on 22 August 2018. Aeolus is one of the Earth Explorer missions proposed by ESA as demonstration paving the way towards measuring wind from space globally (Stoffelen et al., 2005). In this view the continuous effort to better understand and to improve the wind retrieval is undertaken by ESA, Aeolus Data Innovation and

Science Cluster (DISC) and Aeolus CAL/VAL teams since launch. In particular, an increasing effort is undertaken to better understand the various sources of observation errors. Such is a systematic error arising due to a so-called dark current signal anomalies of single Accumulation Charge-Coupled Device (ACCD) pixels (i.e. "hot pixel") (Weiler et al., 2020) or a significant source of wind systematic error which has been found to be correlated with the temperature gradients across the Atmospheric





Laser Doppler Instrument (ALADIN) primary mirror M1 of the telescope (Rennie and Isaksen, 2020). Despite the relatively

large observation errors of the Aeolus wind observations compared to radiosondes or airborne lidar wind observations (e.g. Witschas et al., 2020; Martin et al., 2020) a list of OSEs (Observation System Experiments) provided by various global and regional models showed a significant impact on the NWP[1] such as was suggested by several preelunch studies (e.g. Žagar, 2004; Tan and Andersson, 2005; Weissmann and Cardinali, 2006; Stoffelen et al., 2006; Tan et al., 2007; Marseille et al., 2007, 2008; Weissmann et al., 2012; Horanyi et al., 2015; Šavli et al., 2018). This lead several weather centers[2] to already start with

the operational assimilation of the line-of-sight wind observations.

The main product of the Level-2B processor (L2Bp) is the horizontal line-of-sight (HLOS) wind, which is inferred by the Doppler shift of the backscattered light measured from a small volume in the atmosphere. The backscatter spectrum is sensed by two unique interferometers; the Fabry-Perot (FP) which is used to measure Doppler shift mainly from moving molecules (Rayleigh channel) and the Fizeau which is used to measure Doppler shift mainly from aerosols and small hydrometeors

(Mie channel) (e.g. Reitebuch, 2012). The Doppler shift measured from moving molecules is well described by the so-called Rayleigh-Brillouin spectrum which slightly deviates from the ideal Gaussian spectrum due to atmospheric stratification. Two Brillouin peaks are introduced on each side of the Gaussian spectrum as a consequence of the increased density of molecules lower in the atmosphere (Gu and Ubachs, 2014). The information on temperature and pressure, needed for properly modelling such Rayleigh-Brillouin spectrum characteristics, which should be collocated with the Aeolus measurements, is, however, not

provided by Aeolus, nor available from the current Global Observation System (GOS). Therefor the suggestion of Dabas et al. (2008) was to infer this information from a Numerical Weather Prediction (NWP) which, on the other hand, contains errors.

The mean temperature model error is a well monitored quantity, e.g. it is monitored by the WMO Lead Centre for Deterministic Forecast Verification (WMO-LCDNV[3]). The RMSE of the temperature forecast from various operational global models is found typically to be less than 2 K for a 24h forecast range in the extra-tropics. This error is smaller in the Tropics, being

slightly above 1 K. The majority of global model temperature remains below 3K even for longer forecast ranges. Estimations provided by Dabas et al. (2008) suggest that in a standard atmosphere the temperature sensitivity of HLOS retrieval brings a relative HLOS error of about 0.2 % for 1 K error in temperature, which is well below the 0.7 % relative error specified as an ESA requirement (ESA, 2016). On the other hand, HLOS sensitivity to pressure is suggesting to provide about an order of magnitude smaller impact with typical model pressure errors of few hPa. Compared to the current HLOS observation errors

which are of the order of about $4 \, \mathrm{ms^{-1}}$ (e.g. Martin et al., 2020) it appears to be a rather small overall contribution. However, it is necessary to take into account that NWP model error vary spatially and temporally due to weather regimes present on various spatial and temporal scales that are not always well described. For optimal Rayleigh-Brillouin correction in L2Bp the most accurate source of temperature and pressure information should be used. With the assumption that differences among various model realizations can provide an estimation of the mean characteristics of the model error (i.e. zonally and temporally

---

[1] Aeolus CAL/VAL workshop: https://nikal.eventsair.com/QuickEventWebsitePortal/2nd-aeolus-post-launch-calval-and-science-workshop/aeolus

[2] European Centre for Medium-Range Weather Forecasts (ECMWF) on 9 January 2019, Deutscher Wetterdienst (DWD) on 19 May 2020, Météo-France on 30 June 2020 and Meteorological Office (Met Office) on 8 December 2020

[3] https://apps.ecmwf.int/wmolcdnv/





averaged), the main goal of this paper is to explore the impact of uncertainty in modelled temperature and pressure fields for the retrieval of Rayleigh HLOS winds.

A brief introduction of the Rayleigh-Brillouin correction algorithm is first given in Section 2. Then the methodology of the sensitivity study, is described in more detail in Section 3. Here the production of AUX_MET at Météo-France is described exploiting the local installation of L2Bp. Several statistical validation metrics are as well presented. Results of the sensitivity study are presented in Section 4 following by conclusions in Section 5.

## 2 Description of the Rayleigh-Brillouin correction algorithm

The thorough description of L2Bp algorithm is beyond the scope of this paper and the reader is invited to consult the existing literature (e.g. Tan et al., 2008; Stoffelen et al., 2005) and the L2Bp official documentation (Rennie et al., 2020). A brief overview of the Rayleigh HLOS retrieval from FP interferometer is, however, given to introduce the necessary methodology. The temperature and pressure information is used in a so-called Rayleigh-Brillouin correction algorithm. The information of the spectrometer counts is first used to compute a so-called Rayleigh-Response ($RR$), a quantity that is linearly related to the Doppler shift and hence to the HLOS wind, through a so-called response curve (Dabas et al., 2008). A Doppler shift $\nu_d$ is inferred from a calibration look-up table $\underline{\nu}_d(T, p, RR)$, which specifies the relation between a range of temperatures $T$, pressures $p$, $RR$ and Doppler shift values. This table is provided using the information of backscatter spectrum computed by the Tenti S6 model (Tenti et al., 1974) and the interferometer transmission curves. In particular, for each Aeolus observation the Doppler shift is computed as a linear interpolation specified by Eq. 1.

$$\nu_{d,\overline{cor}}(T, p, RR) = \underline{\nu}_d(T_0, p_0, RR_0) + (T - T_0)\frac{\partial \nu_d}{\partial T}\bigg|_{T_0, p_0, RR_0} + (p - p_0)\frac{\partial \nu_d}{\partial p}\bigg|_{T_0, p_0, RR_0} + (RR - RR_0)\frac{\partial \nu_d}{\partial RR}\bigg|_{T_0, p_0, RR_0}, \quad (1)$$

where subscript 0 refers to nearest values of measured $RR$ and collocated $T$, $p$ available from the $\underline{\nu}_d$ look-up table. Three derivatives are estimated using a finite difference method, as well, from the look-up table.

An additional correction factor is taken into account as the signal from the FP interferometer is contaminated by the Mie signal. The relative contribution of the Mie signal is described by scattering ratio $\rho = 1 + \beta_{aer}/\beta_{mol}$, where $\beta_{aer}$ and $\beta_{mol}$ are particular and molecular backscatter ratio of the sensed volume of atmosphere. A tunable scattering ratio threshold parameter ($\rho_t$) is defined in the L2Bp (De Kloe et al., 2020) which further classifies in to so-called "clear" and "cloudy" wind observations, thus Rayleigh-clear for which scattering ratio is smaller than $\rho_t$ and Rayleigh-cloudy for larger values of scattering ratio. The value of $\rho_t$ has been carefully tuned since the start of the mission and did vary from values of 1.25 in version v3.00 up to 1.6 in version v3.20. This increase lead to generally improved classification (e.g. Rennie and Isaksen, 2020). For any value of scattering ratio larger than 1, an additional correction is applied on top of $\nu_{d,\overline{cor}}$ as specified in the following equation:

$$\nu_{d,cor}(T, p, RR, \rho) = \nu_{d,\overline{cor}}(T, p, RR) + (1 - \rho)\frac{\partial \nu_d}{\partial \rho}\bigg|_{T, p, RR, \rho=1}. \quad (2)$$





Taking into account the relation in between line-of-sight wind LOS and Doppler shift, $\mathrm{LOS} = -\nu_{d,cor}\lambda_0/2$ and $\lambda_0$ the
lidar base wavelength, the sensitivity parameters $\partial_x\mathrm{HLOS} = \partial\mathrm{HLOS}/\partial X$ are provided, where $X$ is $T$, $p$, $RR$ or $\rho$. These are
primarily used to estimate the Rayleigh HLOS wind observation instrumental error internally in the L2Bp, i.e. by assuming
a typical error in temperature and pressure and as well as for the scattering ratio. In addition, these sensitivities along with
reference values of $T_{ref}$, $p_{ref}$ and $\rho_{ref}$ ($T$, $p$ and $\rho$ used in Eq. 1-2) are provided as an output of the L2Bp which can be used
for any additional correction of HLOS without the need for running L2Bp, considering the following equation:

$$\Delta\mathrm{HLOS} = \Delta T\partial_T\mathrm{HLOS} + \Delta p\partial_p\mathrm{HLOS} + \Delta\rho\partial_\rho\mathrm{HLOS}\,, \tag{3}$$

where $\Delta X = X - X_{ref}$ for $X$ and $X_{ref}$ being one of HLOS ($\mathrm{HLOS}_{ref}$ is the output of the L2Bp), $T$, $p$ or $\rho$. Only linear
terms are taken into account thus the correction is expected to be valid for small differences in $T$, $p$ and $\rho$. Several additional
correction schemes are applied in L2Bp (Rennie et al., 2020), however, these are not relevant for the present study.

Data needed for the above mentioned correction are provided to L2Bp as a series of auxiliary files (Rennie et al., 2020) by
the Aeolus Payload Data Ground Segment (PDGS). Three required data inputs have to be specified. First, the Aeolus Level-1B
wind vector mode product which is the main input providing the information from the FP and Fizeau interferometers (e.g.
spectrometer counts), geolocation information, calibration informations and error estimates for several variables. This file is
an output of the Level-1B processor. The second one is the auxiliary meteorological data input (further denoted AUX_MET)
which provides the necessary information on atmospheric temperature and pressure as described previously. The operational
AUX_MET data are produced in near-real time by the Level-2 Meteorological Processing Facility (L2/Met PF) which is a part
of PDGS and is hosted by European Centre for Medium-Range Weather Forecasts (ECMWF).The third one is the auxiliary
input file which provides information of the calibration look-up table $\underline{\nu}_d(T, p, RR)$ and additional data used in the Rayleigh-
Brillouin correction algorithm. Several optional additional input data provide information on the Aeolus predicted orbit ground
track geolocation, climatology of lidar ratio and lidar signal calibration constants. Finally, the additional necessary input file
(further denoted AUX_PAR) provides the settings and parameters to control the L2Bp processing.

To validate the sensitivity of HLOS retrievals due to the Rayleigh-Brillouin scattering dependency on atmospheric tempera-
ture and pressure in operational L2Bp, we consider the Action de Recherche Petite Echelle Grande Echelle (ARPEGE) global
NWP model (Courtier et al., 1991) to provide an independent realization of atmospheric temperature and pressure. This model
will be used to provide the associated AUX_MET input files which are first assessed in a statistical inter-comparison with the
default (i.e. operational) AUX_MET temperature and pressure of the ECMWF IFS model. Based on this comparison and on the
sensitivity parameters $\partial_T\mathrm{HLOS}$ and $\partial_p\mathrm{HLOS}$ the characteristics of the HLOS uncertainty due to temperature and pressure dif-
ferences is discussed. The ability of running L2Bp using ARPEGE derived AUX_MET allows to additionally evaluate the value
of the correction specified by Eq. 3 and the Mie-contamination contribution. The validation is undertaken for the Rayleigh-
clear HLOS observations only, as equivalent sensitivity to temperature and pressure is expected for the Rayleigh-cloud HLOS
observations.



## 3 Methodology and data

### 3.1 Production of Level-2B auxiliary meteorological input at Météo-France

The AUX_MET auxiliary file provides profiles of several meteorological variables such as temperature, pressure, specific and relative humidities, wind as well as information on cloud cover and cloud liquid/ice water content) along the Aeolus orbit. As
Aeolus is operating at both nadir (laser is pointing perpendicular towards the earth surface) and off-nadir (laser is tilted 35 deg off nadir), the AUX_MET data are available for both. Currently the L2/Met PF provides meteorological quantities as vertical profiles interpolated spatially and temporarily from a short term IFS forecast having a validity from 6 to 30 hours and produced twice per day. Thus for the off-nadir mode, which is the mode used for wind measurements, the AUX_MET data do not follow the slant path in vertical. This leads to a difference in terms of geolocation between the model vertical profile and the Aeolus
orbit of about 21 km at 30 km altitude. However, such deviation seems acceptable (Rennie et al., 2020) since it is below the typical effective resolution of the model which does not resolve scales below around 100 km (e.g. Marseille and Stoffelen, 2017). The profiles in operational AUX_MET are provided every about 3 seconds along the predicted Aeolus orbit. Given the satellite ground speed of about $7.4 \ \mathrm{kms}^{-1}$, profiles are separated for about 22 km. This operational auxiliary meteorological file is denoted hereafter as $\mathrm{AUX\_MET_{oper}}$.

The production of auxiliary meteorological data at Météo-France (named $\mathrm{AUX\_MET_{mf}}$) is an off-line procedure that allows to provide data when needed (i.e. specifically for the study described in this paper). In that respect the production of meteorological auxiliary files did not follow exactly the production at L2/Met PF, but has been instead simplified allowing for minimal adaptation of existing operational ARPEGE routines at the time of the study.

A flowchart of the working processing chain is shown in Fig. 1. It is important to notice that a production of $\mathrm{AUX\_MET_{mf}}$
is done every 6 hours, which is consistent with the ARPEGE data assimilation system (Fig. 1b). For a purpose of explaining the behaviour of the chain, a production at 06 UTC is discussed next.

At Météo-France the L1B measurement geolocation is used to provide an information on the Aeolus orbit needed for generating of AUX_MET data. In particular, L1B output files store the geolocation for each measurement, which is ideally available every ∼3 km being a baseline for the sampling distance of vertical profiles in the $\mathrm{AUX\_MET_{mf}}$. Similarly as in
$\mathrm{AUX\_MET_{oper}}$, $\mathrm{AUX\_MET_{mf}}$ vertical profiles are provided both for the nadir and off-nadir, without accounting for the off-nadir slant path geometry. The geolocations are gathered from all available L1B data over the time period of 03 UTC to 09 UTC (typically of about 5-6 L1B files). This is the main input dataset for the production of auxiliary meteorological files.

In the following stage, the extracted geolocation is first reformatted (pre-processing stage in Fig. 1a) in the way compatible with the ARPEGE data assimilation system through BUFR (Binary Universal Format for the Representation of meteorological
data) files. Then, for each geolocation, model vertical profiles of several variables are extracted during the data assimilation window. These ARPEGE outputs in ODB (Observation Data Base) format are finally reformatted (the post-processing stage in Fig. 1) into a single $\mathrm{AUX\_MET_{mf}}$ file ready to be used in L2Bp. The chosen ARPEGE model configuration corresponds to the operational one (CY43T2). This spectral model has a stretched and tilted horizontal grid with the highest horizontal resolution over France (∼5 km) and the lowest (∼20 km) on the opposite pole (New-Zealand). The model consists of 105

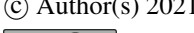



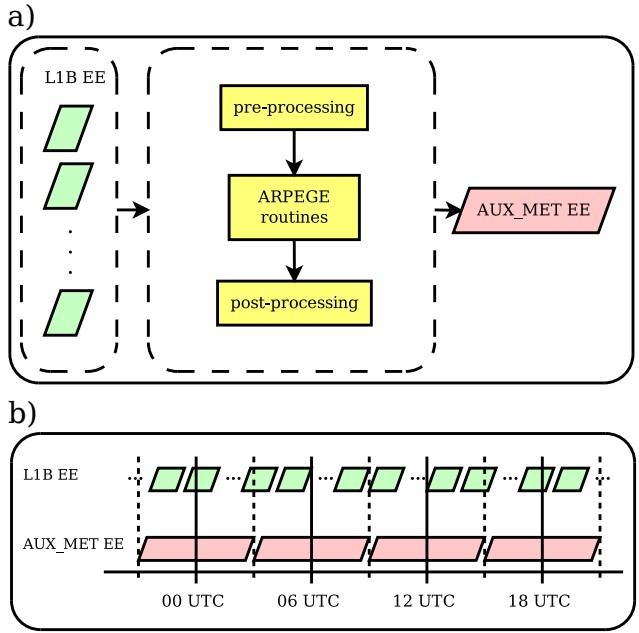

**Figure 1.** Production of AUX_MET files at Météo-France. Detailed description is given in section 3. The (EE) stands for Earth-Explorer file format.

vertical levels defined in between 10 m to 0.1 hPa following a hybrid vertical coordinate. The model profiles are extracted from the ARPEGE model background trajectory in the so-called screening task. This is a part of the 4D-Var system where the model is run forward and compared to observations with 30 min time-slots. For the production of $AUX\_MET_{mf}$ files only part of this task is considered, which is the extraction of model profiles by interpolation of trajectory to the L1B ground geolocation.

   There are several properties of this chain that need to be discussed in more details. The amount of information in the final

$AUX\_MET_{mf}$ is significantly increased compared to $AUX\_MET_{oper}$ files. Indeed, the input to the production chain uses all geolocations extracted from the L1B measurement resolution (that is $\sim$3 km). Thus, compared to the ARPEGE effective resolution, there is a significant amount of redundant data in the $AUX\_MET_{mf}$ files. This is, however, not an issue as L2Bp performs internally a nearest-neighbour interpolation of $AUX\_MET_{mf}$ profiles. One of the main deficiencies of the flowchart presented in Fig. 1 is that $AUX\_MET_{mf}$ files are not overlapping in time as is the case for the $AUX\_MET_{oper}$ files. This is

since the validity of $AUX\_MET_{mf}$ is exactly 6 hours (i.e. it represents the 6-12 hour ARPEGE forecast) and are produced four times per day, whereas $AUX\_MET_{oper}$ represents 6-30 hour IFS forecast and are produced twice per day. When using $AUX\_MET_{mf}$ files in the L2Bp, thus a specific approach must be used when the L1B input file spans the time in between two data assimilation windows of ARPEGE system (i.e. validity of two $AUX\_MET_{mf}$ files). In such a case the L2Bp must be run two times for each of the $AUX\_MET_{mf}$ file. The information from both L2B runs is finally merged into one L2B output. This

is not ideal from operational perspective, which, however, is not in the scope of this study.





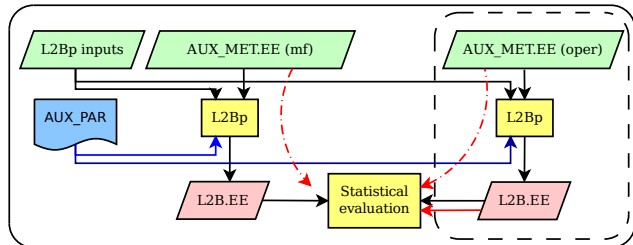

**Figure 2.** Flowchart of L2B generation and sensitivity study set-ups. Detailed description is given in section 3. The (EE) stands for Earth-Explorer file format.

## 3.2 Sensitivity study

From two distinct auxiliary meteorological data ($AUX\_MET_{oper}$ and $AUX\_MET_{mf}$), covering the same Aeolus orbit section, it is possible to analyze differences induced on the two associated L2B HLOS retrievals. The necessary processing chain for performing such a study is displayed in Fig. 2. The part enclosed in the dashed lined box corresponds to the simplified

operational L2B processing chain performed by the PDGS which essentially consists of running the L2Bp with all necessary input files and a configuration file. A similar chain of processes has been replicated locally at Météo-France, using the same L2Bp version, the same configuration file and the same input files, except for the $AUX\_MET_{mf}$ input file which is produced as described in section 3.1. The statistical evaluation of the difference between the two HLOS wind products, to be explained in more details in section 3.4, is shown in Fig. 2 by the two black arrows pointing towards the "statistical evaluation" box.

The another evaluation approach is used as is depicted by the three red arrows in Fig. 2. In this particular case the L2Bp is not used but instead all the necessary information is directly extracted from the $AUX\_MET_{oper}$ and $AUX\_MET_{mf}$ files and the operational L2B product. As described in section 2 given the small model differences in terms of temperature and pressure with respect to reference values (from IFS) it is possible to express the HLOS difference (Eq. 3) as a linear correction function. It is first assumed that the third term in Eq. 3 can be neglected, i.e. $\Delta\rho = 0$. Then, the computation of temperature and pressure

differences (i.e. $\Delta T$ and $\Delta p$ in Eq. 3) is done by comparing the two AUX_MET files. This has been possible by closely replicating the L2Bp algorithm that first interpolates temperature and pressure (nearest profile from the AUX_MET) onto the HLOS measurement geolocation. Note that HLOS *measurement* is representative for $\sim 2.8$ km, whereas HLOS *observation* represents the accumulation of up to about 30 (for Rayleigh-clear) of these measurements considering a weighted average (currently performed with equal weights). The same weighted averaging applies for temperature and pressure information

representative of a particular HLOS observation. In this second approach the temperature and pressure values collocated with a particular HLOS observation are computed separately for both $AUX\_MET_{oper}$ and $AUX\_MET_{mf}$. It is important to note that for the $AUX\_MET_{oper}$ the reference temperature and pressure values are already available from the L2Bp output file along with all other meta-parameters for each HLOS observation. However, to reduce the error of the described (simplified) algorithm of computation of temperature and pressure at the HLOS observation level, they have been used only for an additional quality-





control but not for the sensitivity study itself. Finally, using the sensitivity values ($\partial_T$HLOS and $\partial_p$HLOS) available in the operational L2B output the HLOS difference has been computed.

### 3.3  Datasets and Level-2B processing

As Aeolus is an explorer mission the retrieval of HLOS winds from raw satellite data is continuously adapting and improving. The choice of selected dataset is therefor an important factor that should be taken into a consideration. For this particular study

the dataset is part of the reprocessed one available for a period of July-December 2019 (i.e. baseline 2B10). This is an official reprocessing from PDGS and contains a number of improvements of several deficiencies closely examined by ESA and DISC. This is an improved treatment for removing spurious outliers induced by dark current signals (hot pixels) (Weiler et al., 2020). A dark current signal is measured when no laser pulse is emitted. Moreover, orbit variable systematic errors have been removed by an innovative method based on a linear relation of the satellite primary mirror temperature gradient and the model estimated

HLOS systematic error (Rennie and Isaksen, 2020). In addition, since 2019-08-01 the Rayleigh-Brillouin correction has been affected by calibration update, having a direct impact on the properties studied in this paper. As a consequence, the dataset selected for this study is valid for the period of 2019-08-01 up to 2019-12-31. This dataset is used for the study, concerning the methodology of a second approach of the sensitivity validation (without using the L2Bp) as described in section 3.2. On the other hand, data from the first Aeolus laser period (so-called FM-A period) is used for the experiment using L2Bp. In this

case the dataset consists of 464 orbits from the period of 2018-11-30 to 2019-01-13. As this particular dataset has not been yet officially corrected it can impact results presented afterwards, however, as will be discussed later this effect is small.

The L2Bp used here is of version 3.01. The configuration file of the processor (AUX_PAR) is of version 8 (used during baseline 2B02), which consists of a scattering ratio threshold of 1.5-1.6 for Rayleigh classification on clear and cloudy scenarios and a Rayleigh-clear accumulation length (i.e. the distance along the orbit for which the observation is representative) of 89

km.

### 3.4  Statistical evaluation

From L2B outputs only Rayleigh-clear information is used after applying a basic quality-control. It consists of rejecting all observations identified as invalid (information available in the L2B output) or for which the L2B estimated observation error is larger than $10\ \mathrm{ms^{-1}}$ (similar to a range of values typically used in the QC of several NWP centers assimilating Aeolus data).

Such approach allows to study the sensitivity properties only for data that would be considered for data assimilation in NWP models.

Main properties of $\Delta$HLOS, $\Delta T$ and $\Delta p$ are examined by using typical statistical metrics. Such are $\mathrm{median}$ (50th percentile of the distribution) and median absolute difference ($\mathrm{mad}$) evaluated spatially and/or temporally, along with several other distribution percentiles. These quantities are especially useful in case of outliers in the distribution. The so-called $\mathrm{madn} =$

$1.48\mathrm{mad}$ is used instead of $\mathrm{mad}$ since it agrees with the standard deviation when the sample distribution is Gaussian. The same approach is chosen to describe the statistical properties of the HLOS sensitivity terms, e.g. $\partial_T$HLOS.



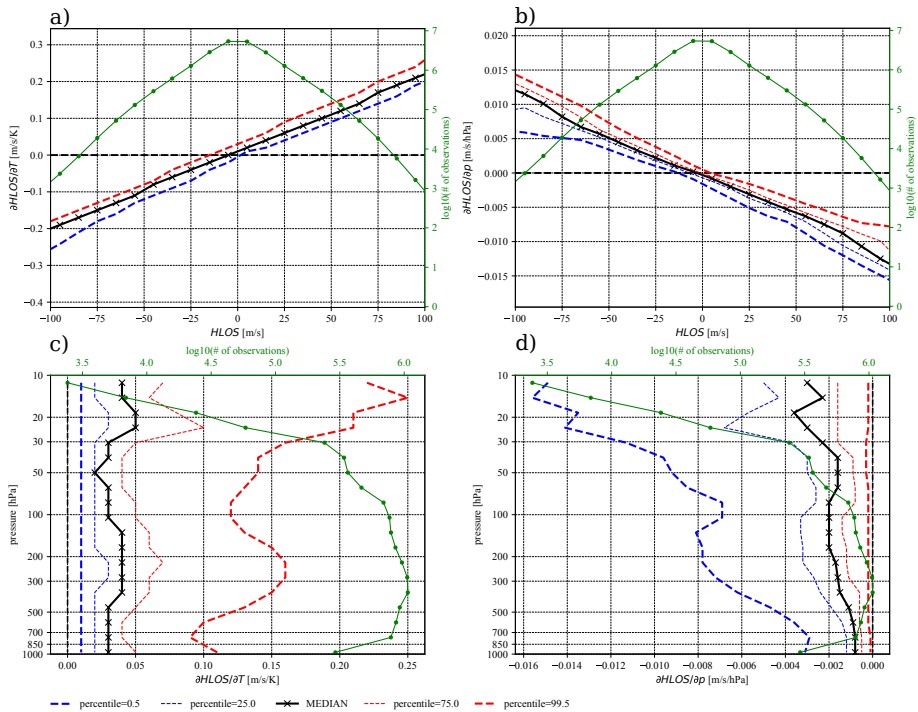

**Figure 3.** The HLOS sensitivity (a, c) with respect to temperature (i.e. $\partial_T$HLOS) and (b, d) with respect to pressure (i.e. $\partial_p$HLOS) over a 5-month period (2019-08-01 to 2019-12-31). (a, b) The sensitivity is shown as a function of HLOS wind velocity. The median and several percentiles are computed by gathering all available and valid data in HLOS bins of $10 \, \mathrm{ms}^{-1}$. (c, d) The sensitivity is shown as a function of pressure taking into account only data with positive HLOS wind velocity. (solid-green) The amount of available data in a bin.

# 4   Results

## 4.1   Main characteristics of the HLOS sensitivity to temperature and pressure

Main properties of the sensitivity (i.e. $\partial_T$HLOS and $\partial p$HLOS) valid for the period from 2019-08-01 up to 2019-12-31 are
shown in Fig. 3. Statistics are displayed in Fig. 3(a, b) as a function of HLOS wind velocity. These are computed by grouping available data in to $10 \, \mathrm{ms}^{-1}$ HLOS wind velocity bins. In addition to the average sensitivity, measured by the median the percentiles of 0.5 and 99.5 (as well 25 and 75 for the sensitivity against pressure) are also computed to give an estimate of its variability for a particular range of HLOS wind velocities.

One property of the HLOS sensitivity to temperature is its approximate linear dependency with HLOS wind velocity. The
slope of the median curve is $\sim 0.002 \, \mathrm{K}^{-1}$ which is in good agreement with the value for the standard atmosphere given by Dabas et al. (2008). The variability shown by the two percentiles is mostly induced by the fact that inside each bin HLOS wind velocity varies by $10 \, \mathrm{ms}^{-1}$ and is less effected by the fact that atmospheric conditions, i.e. temperature and pressure, vary significantly over the same range bin. The variability is rather consistent over the whole range of HLOS wind velocity.





As discussed in section 2, by using properties of the sensitivity presented in Fig. 3a it is possible to approximately estimate
the random error contribution since temperature information is not known exactly (i.e. based on Eq. 3). Typically, for a random
error of 1 K in temperature the random error in HLOS would increase by only of about $0.2\,\mathrm{ms}^{-1}$ for a $100\,\mathrm{ms}^{-1}$ HLOS wind.
However, for temperature differences of about 10 K, up to about $2\,\mathrm{ms}^{-1}$ increase in HLOS (for $100\,\mathrm{ms}^{-1}$ HLOS wind) is
expected, i.e. a relative error of 2 %. As a consequence, outliers in the model temperature distribution are going to produce
significant differences in HLOS retrievals.

Moreover, sensitivity varies in the vertical, as shown in Fig. 3c, as a function of pressure (i.e. altitude) but only for positive
HLOS winds since it is symmetric for negative HLOS. The largest values are expected near the upper-troposphere-lower-
stratosphere (UTLS) both of which are associated in average with globally stronger winds. Similarly, the sensitivity increases
with altitude in the stratosphere due to the positive temperature gradient in this region.

On the other hand, the HLOS sensitivity to pressure is significantly less, given the small uncertainties in model pressure
information. In particular, the sensitivity shown in Fig. 3(b, d) indicates a median value (over all positive HLOS) of about
$-0.002\,\mathrm{ms}^{-1}\mathrm{hPa}^{-1}$ which is in range of expected values (e.g. $0.003\,\mathrm{ms}^{-1}\mathrm{hPa}^{-1}$ for the standard atmosphere as shown by
Dabas et al. (2008)). The variability of the sensitivity (Fig. 3b) becomes stronger with larger absolute HLOS winds, which
is induced by the effect of variable temperature and pressure conditions in a given HLOS bin. Therefore, a significant slope
(especially for percentile 0.5) of the sensitivity with altitude is shown in Fig. 3d. With a 10 hPa (a typical root-mean-squared
error of 24 hour global model mean-sea-level forecast against analysis[4]) error in pressure, the random error in HLOS would
increase by about 0.1 %, whereas a 100 hPa random error would increase the HLOS error by 1 %. The effect of the pressure
sensitivity is thus expected to be at least one order of magnitude smaller compared to the temperature sensitivity when the
NWP model pressure is provided in AUX_MET.

Overall the statistics shown in Fig. 3 is very consistent over the chosen time period of 5 months, i.e. no significant variations
have been observed when shorter time periods have been analyzed separately (not shown). The bulk analysis of the sensitivity,
as presented above, is therefore sufficient.

## 4.2 Evaluation of model temperature and pressure uncertainty

The comparison of the $\mathrm{AUX\_MET_{oper}}$ and $\mathrm{AUX\_MET_{mf}}$ files allows to estimate mean uncertainties in model pressure and
temperature with their effects on HLOS sensitivities. For that purpose a second approach described in section 3.2 has been
proposed. The temperature and pressure differences between ARPEGE and IFS (i.e. $\Delta T$ and $\Delta p$) have been computed from
AUX_MET by mimicking the accumulation algorithm in L2Bp.

The average difference is less than about 0.5 K in temperature and 0.25 hPa in pressure below 30 km altitude (Fig. 4),
showing an overall good consistency among the two model atmospheric temperature and pressure averaged globally and in
time. The difference in temperature is in average the largest over a pressure layer of 100-300 hPa, mainly over both poles (see
Fig. 5c). In the tropics the mean difference is largest above and below the tropical stratosphere near the minimum temperature
core (see Fig. 5a) and higher in stratosphere. Larger average difference is also noticed near the surface in polar regions. For this

---

[4]https://apps.ecmwf.int/wmolcdnv/scores/mean/msl





particular dataset the IFS zonal mean temperature and its variability are shown in Fig. 5(a and b). These are broadly consistent with available reanalysis statistics (i.e. Hersbach et al., 2020), taking into account the rather short period of the current dataset (5 months). Namely, the largest temperatures are found in the tropical troposphere and the lowest temperatures are observed

in UTLS. The amount of data used for statistics (Fig. 5e) becomes reduced especially above about 10-15 hPa, where they become unreliable. The largest temperature zonal variability (Fig. 5b) over the 5 month period is found over the south pole which coincides with the lowest zonal mean temperature (Fig. 5a) area in the stratosphere. This is a typical feature of the Southern Hemisphere winter period (e.g. Matsushita et al., 2020). The ARPEGE model has a higher polar tropopauze and a slightly reduced temperature gradient in tropical stratosphere, when compared to the IFS model.

A good consistency between model temperature fields is not only observed for the mean distribution. For 50 % of the distribution the differences remain below 1 K and 0.5 hPa almost everywhere below 30 km altitude (Fig. 4). For temperature this uncertainty is rather similar at all altitudes, although it is slightly larger in the stratosphere. A noticeable variability in temperature difference exists in the tropical stratosphere as shown in Fig. 5(d). The largest amplitude in temperature difference, although spatially very localized, is observed near the ground over the two poles (Fig. 5d). Given that the chosen dataset

describes primarily the southern hemisphere winter months slightly increased variability is as well observed high in the stratosphere over the south pole (Fig. 5d). This is consistent with the largest variability in IFS model temperature observed over that area as shown in Fig. 5b. These regions, therefore, represent the most sensitive (in respect of temperature) areas for the HLOS retrieval in the studied period.

A scenario displayed in Fig. 7c presents typical differences of model temperature fields. Here data (each coloured box

represents a valid observation) is extracted from a section of the Aeolus orbit crossing the regions of South Indian Ocean, tropical Indian Ocean and Southeast Asia. As shown in Fig. 7b several deep cloud systems exist on the path of Aeolus, increasing the lidar signal attenuation, as denoted by increased scattering ratio. In such situations temperature differences are found fairly consistent with the overall statistics shown in Fig. 5c, this is an overall positive $\Delta T$ along with vertical layers where $\Delta T$ becomes negative (e.g. especially above the tropical troposphere). A significant amount of granulation in Fig. 7c

accounts for the model error. Only few events of $|\Delta T|$ larger than 1-2 K is observed.

The largest deviations between the two NWP models can be further identified by analyzing outliers of the difference probability density function (PDF) better shown in Fig. 4. Tails of the PDF are presented by the percentiles 0.5 and 99.5 which includes 99 % of data. It can be seen that model temperature difference will exceed about 2 K in the mid-troposphere and 3-4 K near the ground and in higher layers of troposphere and stratosphere for only 1 % of the distribution. For pressure this goes to

about 1.5 hPa near the ground and decreases with altitude. In less than 1 % of cases, differences are even larger, although, the overall summary of the results shown in Fig. 4 is that differences in temperature and pressure between the models are overall small.

Model differences properties displayed in Figures 4-5 and sensitivities shown in Fig. 3 can be combined to estimate largest expected HLOS variations due to the Rayleigh-Brillouin effect. The largest variations in HLOS are expected for temperature

particularly in the stratosphere, due to larger HLOS sensitivity, but also near the ground, due to significant temperature differences between the two models. For an HLOS value of about 100 ms$^{-1}$ the HLOS variations could exceed 0.6 ms$^{-1}$ in less



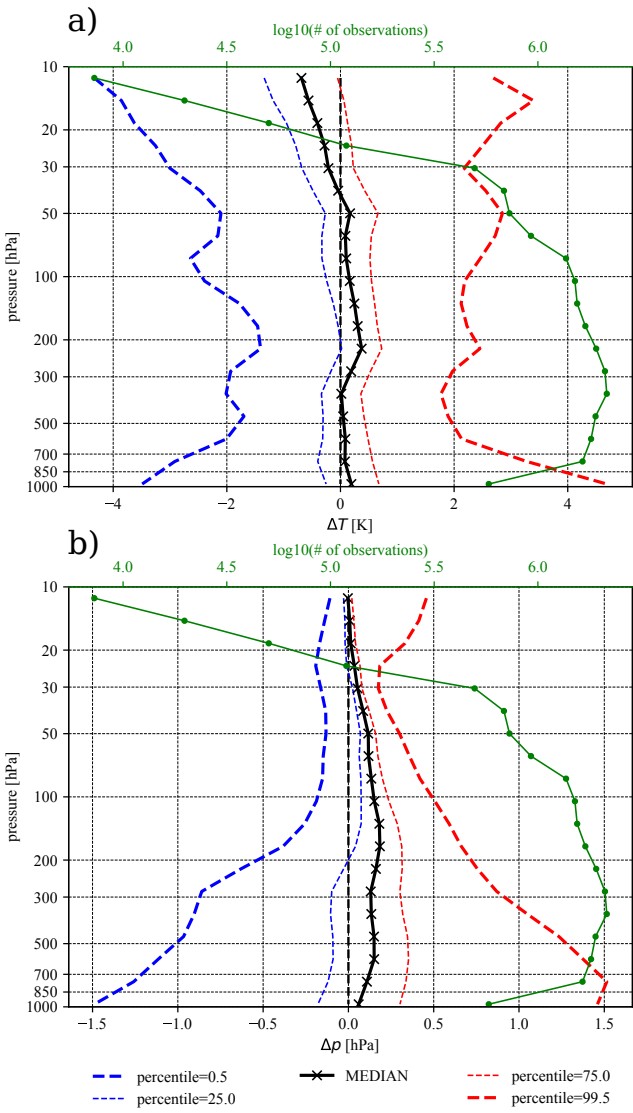

**Figure 4.** The (a) temperature and (b) pressure difference statistics between ARPEGE and IFS produced for the same period as in Fig. 3 and with all valid data available globally along the Aeolus orbits. Temperature and pressure values are computed as in L2Bp, i.e. weighted average over the accumulation of Rayleigh-clear observations.

than 1 % of the situations. On the other hand the maximum expected variation of HLOS due to differences in pressure fields could be about 0.015 ms$^{-1}$ at HLOS of 100 ms$^{-1}$ but only near the ground (which is very unlikely) in less than 1 % of the cases. Therefore the pressure component has no significant impact on the HLOS sensitivity and will not be analyzed further.



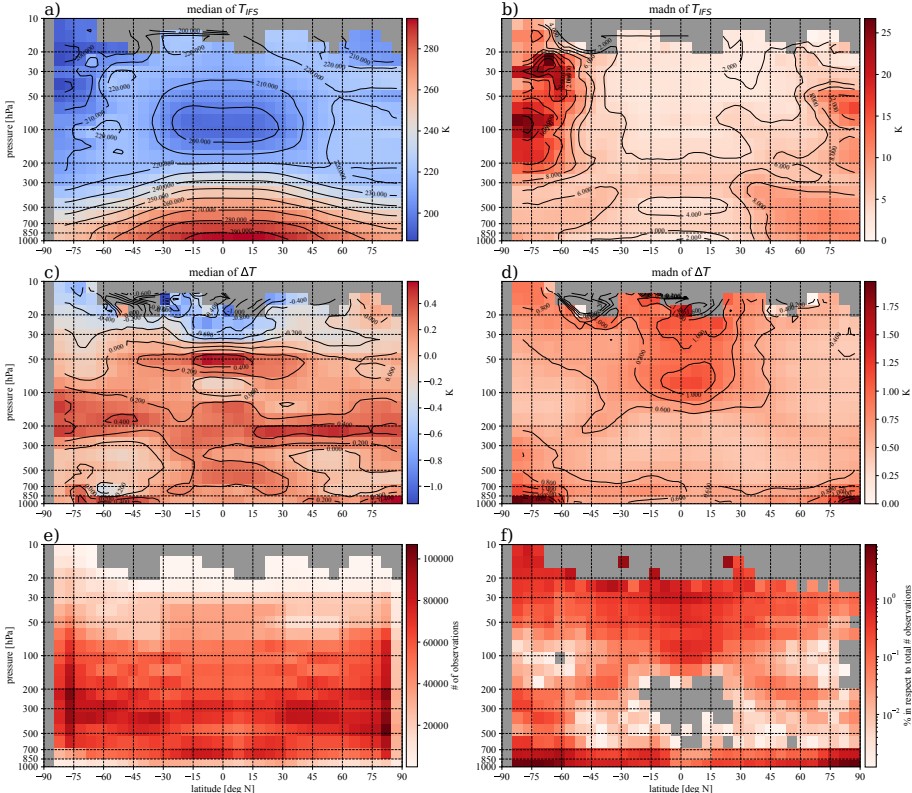

**Figure 5.** The zonal and temporal statistics of the IFS temperature (a and b) and temperature differences between ARPEGE and IFS (c and d) computed for the same period as in Fig. 3-4. (a and c) The median and (b and d) madn $= 1.48$mad are shown. In addition (e) shows the associated number of data for the statistics and (f) the number of samples found with a temperature difference $\Delta T > 4$ K used in the discussion of the Fig. 8.

## 4.3 Level-2B HLOS derived by using ARPEGE model temperature

By combining the HLOS sensitivity of the Rayleigh interferometer for Aeolus (section 4.1) with temperature differences between ARPEGE and IFS models (section 4.2), the HLOS correction ($\Delta$HLOS as defined in Eq. 3) can be next evaluated.

Similar statistics as for $\Delta T$ are computed for $\Delta$HLOS (Fig. 6). The spatial and temporal statistics lead to a symmetric PDF as shown in Fig. 6a, with madn larger than about 0.2 ms$^{-1}$ above 30 hPa. The curve for the madn almost overlaps with the 75th percentile curve, which along with the 25th percentile presents 50 % of the distribution. Since a sensitivity $\partial_T$HLOS strongly depends on HLOS wind amplitude (Fig. 3) statistics for $\Delta$HLOS should display this dependency. This is confirmed by Fig. 8a. Here the median is slightly larger only for HLOS winds larger than about 100 ms$^{-1}$. On the other hand, several outliers with significantly larger values (up to more than 5 ms$^{-1}$ - not shown) of $\Delta$HLOS are observed. However, these outliers do not significantly affect on the statistical properties of the $\Delta$HLOS, in particular, 99.9 % of the distribution is still confined within the 1 % HLOS slope.





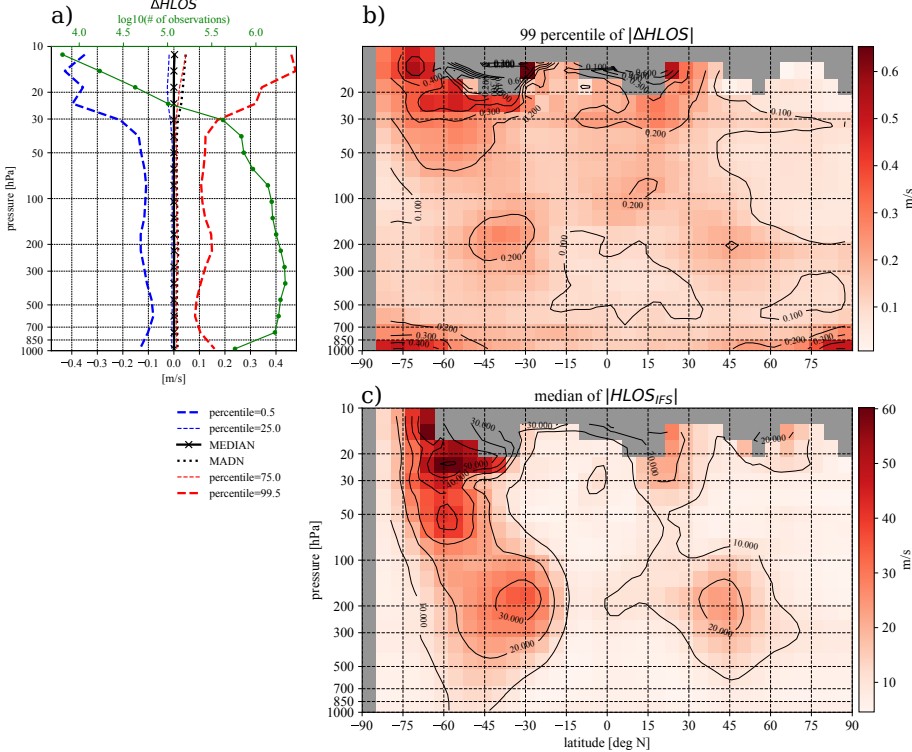

**Figure 6.** The $\Delta$HLOS statistics related to the dateset presented in Fig. 3-5 and computed by Eq. 3. (a) The spatial and temporal statistics are displayed as a function of pressure showing the median, madn and several percentiles. (solid green) The number of available data at specific pressure layers. (b) 99th percentile of $|\Delta$HLOS$|$ and (c) the median of the $|$HLOS$_{ifs}|$ from the operational L2B output files.

The largest difference (i.e. 1 % of the PDF) in Fig. 6a exceeds $0.1$ ms$^{-1}$ below about 30 hPa and increases to about 0.4 ms$^{-1}$ above. Although temperature differences are in 99 % of the time smaller than about 4 K (Fig. 4a), the remaining 1 % of the distribution introduces differences of several 10 K. To better distinguish such outliers from the remaining distribution, statistics is reevaluated for events with a temperature difference lower than 4 K (Fig. 8b). This simple filter efficiently removes

all situations for which a relative difference $\Delta$HLOS/HLOS $> 1$ %. The comparison between Fig. 8a and Fig. 8b reveals that the largest differences in HLOS are associated with the outliers of the temperature difference $\Delta T$ distribution. Fig. 8b shows that over the whole range of HLOS values the difference $\Delta$HLOS is well confined below a 1 % slope of HLOS, and that in 99 % of situations it is well confined below a 0.7 % slope (i.e. ESA mission requirements).

The remaining question concerns the main characteristics of situations with $\Delta T > 4$ K. This can be discussed first by

examining where such cases typically take place. Fig. 5f is displaying the percentage of such cases relative to the total number of cases found in each latitude-pressure bin (i.e. Fig 5e). The percentage shown in Fig. 5f is closely related to the madn of the $\Delta T$ shown in Fig. 5d. The largest contribution of the outliers (up to 10 % locally) is thus observed near the ground on both poles. A relatively large contribution is also noticeable in tropical stratosphere, south pole stratosphere (above $\sim$50 hPa) and

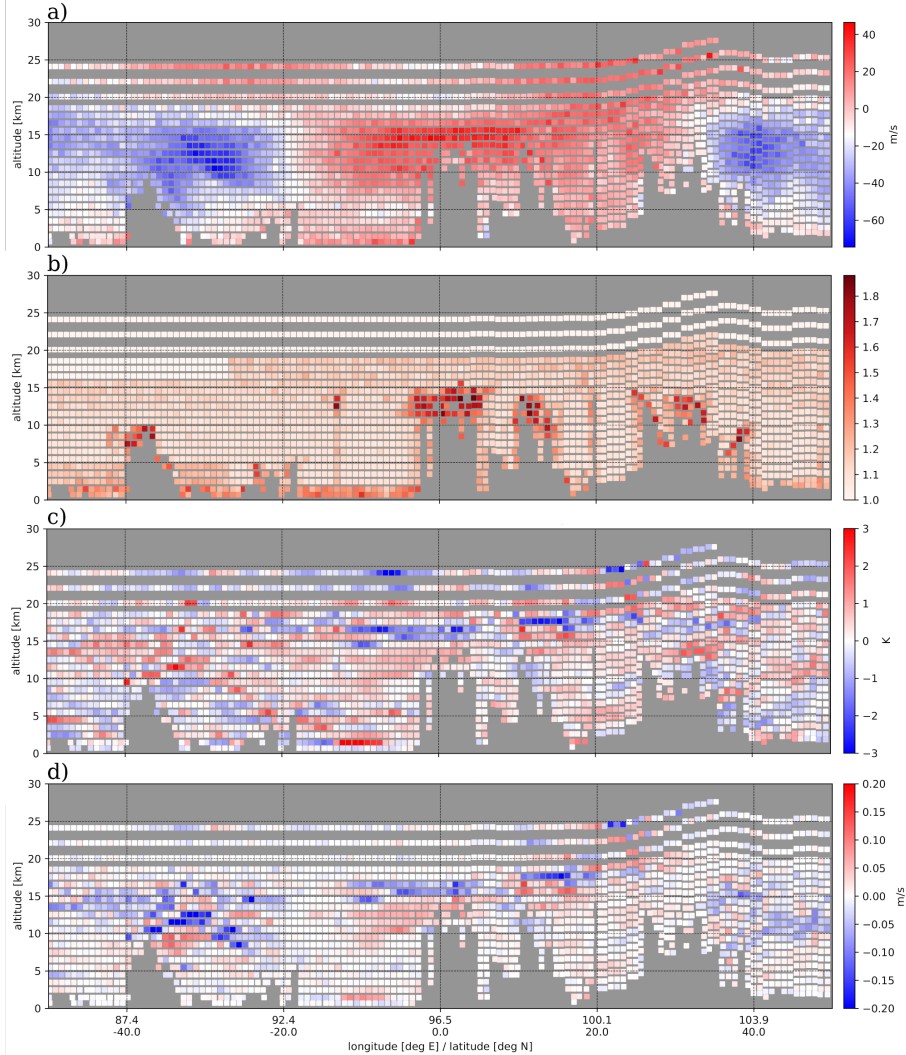

**Figure 7.** Vertical cross-section of an Aeolus orbit extending over the area of South Indian Ocean, tropical Indian Ocean and Southeast Asia on 13th September 2019 at about 23 hour UTC. (a) A $\mathrm{HLOS_{ifs}}$ wind amplitude, (b) scattering ratio, (c) $\Delta T$, i.e. a temperature difference between ARPEGE and IFS and the resulting (d) $\Delta \mathrm{HLOS}$ as specified by Eq. 3. Note: dimensions of coloured boxes are symbolic, hence, not representing the area of representativeness of observations.

as well around 200 hPa at latitudes 30-40 N/S. The later is closely correlated with the position of the subtropical jet-stream as

can be confirmed by the median of IFS HLOS in Fig. 6c.

The largest values of $\Delta \mathrm{HLOS}$ are produced by two mechanisms as is revealed by the outliers of the distribution displayed in Fig. 6b. First, the difference in HLOS is larger for in absolute larger HLOS winds, as is a consequence of the linear relation of $\partial_T \mathrm{HLOS}$ presented in Fig. 3a. This effect is confirmed by comparing it with the average HLOS shown in Fig. 6c. Thus, the





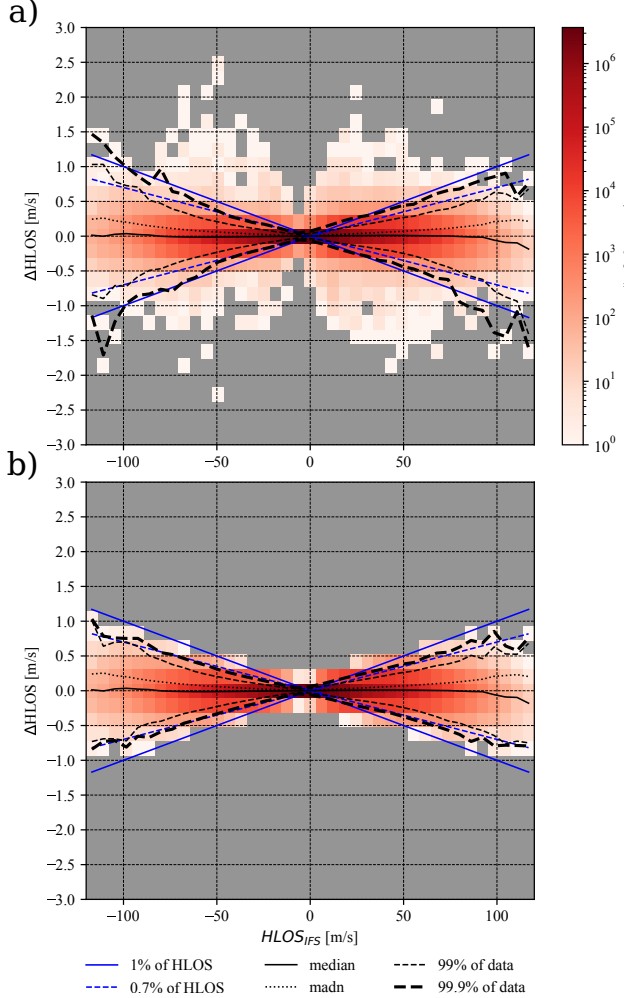

**Figure 8.** (a) Statistics of L2B HLOS differences $\Delta\text{HLOS} = \text{HLOS}_{\text{ARPEGE}} - \text{HLOS}_{\text{IFS}}$ as a function of $\text{HLOS}_{\text{IFS}}$ wind. (b) Similar statistics but including only events associated with a temperature difference $\Delta T = T_{\text{ARPEGE}} - T_{\text{IFS}}$ lower than 4 K in absolute values. Color shading is associated with the density of events in a particular x-y area, with gray color showing when data are not available. Several statistics are provided for a range of HLOS winds such as (solid black) median and (dotted black) madn. The two percentiles (dashed black) indicate the range of values included in 99 % (or 99.9 %) of the $\Delta\text{HLOS}$ distribution for a particular HLOS value, highlighting outliers. (blue) The 1 % (and 0.7 %) HLOS slopes used as a reference in the discussion.

largest differences in HLOS are expected near the location of subtropical jet-streams. Due to the southern hemisphere winter

months prevailing in the dataset the stratosphere polar night jet is clearly evident over the south pole (Fig. 6c). The second effect producing large values comes from the largest temperature differences (i.e. Fig. 5f). This contribution is strongest near the ground over both poles but it also contributes to increased HLOS differences in the tropical stratosphere and over the


subtropical jet-streams. Overall the largest relative differences $\Delta$HLOS/HLOS are expected near the ground over both poles where the HLOS is in average small in absolute values.

The effect of, mainly, the first mechanism is well observed in the scenario displayed in Fig. 7d, taking into account associated temperature differences (Fig. 7c) and HLOS amplitude (Fig. 7a). Largest increments $\Delta$HLOS are observed over the subtropical jet-stream regions (especially in the South Hemisphere) and as well in the tropical UTLS. In particular situation the largest differences in HLOS are found near the active deep cloud systems.

### 4.4    The contribution of the Mie contamination

The advantage of running L2Bp is that it allows to cross-validate Eq. 3 and thus to estimate the relative importance of the Mie contamination sensitivity term (i.e. in Eq. 2). To examine this contribution a second dataset is used (the FM-A period) to first produce the $\text{AUX\_MET}_{\text{oper}}$ and $\text{AUX\_MET}_{\text{mf}}$ files and then L2Bp is run to generate $\text{HLOS}_{\text{oper}}$ and $\text{HLOS}_{\text{mf}}$ as described in section 3, respectively. In addition, $\text{HLOS}_{\text{mf,estimate}} = \text{HLOS}_{\text{oper}} + \Delta\text{HLOS} = \text{HLOS}_{\text{oper}} + \Delta T \partial_T \text{HLOS}|_{\text{oper}} + \Delta p \partial_p \text{HLOS}|_{\text{oper}}$ is computed using Eq. 3, with metadata provided from the L2B output of $\text{HLOS}_{\text{oper}}$ and $\Delta T$, $\Delta p$ from the
two AUX_MET files. Taking into account Eq. 1-2 and $v_{LOS} = -\nu_d\lambda_0/2$, the following equation can be derived.

$$\text{HLOS}_{\text{mf,estimate}} - \text{HLOS}_{\text{mf}} \approx -(1-\rho)\left(\partial_\rho\text{HLOS}|_{\text{mf}} - \partial_\rho\text{HLOS}|_{\text{oper}}\right) = -\Delta \tag{4}$$

where, subscripts mf and oper essentially define conditions of $T$, $p$, $\rho$ and $RR$ at which derivatives are computed. So, the difference between the methodology of Eq. 3 and L2Bp allows to estimate properties of the contribution of the Mie contamination due to uncertain temperature and pressure information that effects the HLOS scattering ratio sensitivity term.

Since the FM-A dataset used here has not yet been officially reprocessed, it was first necessary to evaluate the sensitivity $\partial_T\text{HLOS}|_{\text{oper}}$. Its properties have been found (not shown) very similar and consistent with the one from the FM-B dataset (i.e. 3a). In particular the slope of $\sim 0.002$ K$^{-1}$ is evident, although, overall the sensitivity in FM-A dataset is for a value of about $0.05$ ms$^{-1}$K$^{-1}$ smaller. Regarding the results presented further this has been found insignificant.

The $\Delta$ value has been found overall small as expected. Its PDF distribution is symmetric with values smaller than $0.05$
ms$^{-1}$ for 99 % of cases in absolute value for the dataset of interest. This suggests that the computation of correction using Eq. 3 compares well with the direct use of L2Bp in the particular case with the two AUX_MET files. This result suggests that in practice it is preferable to compute the HLOS correction using Eq. 3 when small differences in model temperature and pressure are expected, since it is faster and technically less demanding than rerunning the L2Bp using different AUX_MET file. On the other hand, a portion of the distribution of $\Delta$ (i.e. 1 %) have been found having values of $0.4$ ms$^{-1}$ in absolute.
These differences are a consequence of differences that arise in the HLOS sensitivity to scattering ratio evaluated with different atmospheric situations (i.e. temperature and pressure) as reflected by Eq. 4.

The $\Delta$ values are in average smaller than the HLOS differences due to temperature and pressure (i.e. $\Delta$HLOS estimated by Eq. 3) as shown in Fig. 9. In average the HLOS sensitivity to scattering ratio brings additional 10-20 % to HLOS correction when $\Delta$HLOS $> 0.2$ ms$^{-1}$. The larger the value of $\Delta$HLOS is the smaller the relative contribution becomes. However for





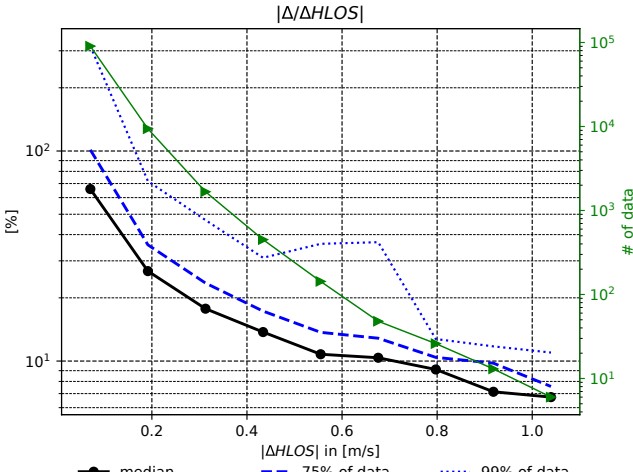

**Figure 9.** The relative contribution to the HLOS total correction of the HLOS sensitivity to the scattering ratio ($\Delta$) compared to the contribution of the HLOS sensitivity to temperature and pressure ($\Delta$HLOS). A complete description is given in Section 4.4. For each value on the x-axes (bin spacing of $0.12 \, \mathrm{ms}^{-1}$) statistics (median, 75th and 99th percentile) are presented together with the amount of data available (green solid line).

large values of $\Delta$HLOS, the amount of data used in the statistics becomes small (green line in figure), so the estimation is not significant, but this decreasing trend with increasing $\Delta$HLOS is apparent. When $\Delta$HLOS is small value of $\Delta$ can increase up to about 70 % for the distribution median or even more than 100 % in about 1 % of cases. However, for small $\Delta$HLOS values the overall correction to the HLOS correction is not of significant practical importance. Overall the sensitivity to scattering ratio is of less importance, even for relatively large scatter ratio threshold of 1.6 used in the L2Bp, compared to the sensitivity to temperature.

## 5 Discussion and Conclusions

We conducted the sensitivity study on the Aeolus HLOS wind product to atmospheric temperature and pressure fields using the operational Aeolus data, ARPEGE model temperature and pressure short-term forecasts and the Level-2B processor (L2Bp). The main goal was to evaluate the impact of the uncertainty in modelled temperature and pressure fields on the Rayleigh-Brillouin correction of the HLOS wind retrieval in L2Bp mainly from a perspective of data assimilation of HLOS winds in NWP. The two methods have been proposed. The first one estimates the possible HLOS correction through metadata provided by the operational L2Bp output files and the second one consists of running the L2Bp locally at Météo-France.

The main assumption of the study has been that basic overall (e.g. time and zonally averaged) characteristics of NWP model temperature and pressure uncertainties can be estimated by comparing atmospheric fields from two different global models (ARPEGE and IFS). For the L2B retrieval these uncertainties are propagated through the Rayleigh-Brillouin correction algorithm and are also affecting the Mie-contamination process. They reflect on the HLOS amplitude as an additional source





to the random observation error. In comparison to the known Aeolus Rayleigh-clear HLOS observation errors, which is about $4 \, \mathrm{ms}^{-1}$ (Martin et al., 2020), the uncertainty due to the Rayleigh-Brillouin correction represents only a small contribution. The main characteristics of this additional HLOS error distribution is that in 99 % of cases the HLOS amplitude differs by only 1

% leading to a maximal difference of about $1 \, \mathrm{ms}^{-1}$ for an HLOS wind value of $100 \, \mathrm{ms}^{-1}$. On the other hand, in about 1 % of the cases the HLOS retrieval will differ by up to about 3 % with few cases having differences of several $\mathrm{ms}^{-1}$. These outliers are associated with temperature differences larger than about 4 K. The sensitivity to pressure is at least an order of magnitude less important and was not studied in details. The overall small absolute differences in HLOS amplitude, which are in 99 % of cases less than $0.15 \, \mathrm{ms}^{-1}$ and mostly located below $30 \, \mathrm{hPa}$, coming from model temperature and pressure uncertainties are in

good agreement with expected values from the study of Dabas et al. (2008).

    The largest differences in temperature between the two models over a 6 month period have been noticed over the poles near the surface which is believed to at least partly reflect the differences in model orography as the ARPEGE model has a stretched horizontal grid with the highest resolution over Europe. On the other hand, differences in the lower to upper stratosphere and near the subtropical jet are more realistically reflecting on the model temperature errors linked to dynamical

and physical processes. The model error patterns are found to be fairly realistic, e.g. when compared to spread of the ECMWF Ensemble of Data Assimilation (EDA) system as presented by Isaksen et al. (2010) (their Fig. 5a), comforting our initial assumption. However, their amplitudes might be underestimated. Despite this limitation results showed the main impact of model temperature errors in the L2Bp on HLOS wind retrievals.

    By running the L2Bp using different temperature and pressure input information it has been possible to study the impact

of the uncertainty in modeled temperature and pressure on Mie-contamination HLOS correction. This correction is applied after the Rayleigh-Brillouin correction in the L2Bp. In particular the sensitivity of the HLOS to the scattering ratio reveals that the HLOS retrieval algorithm slightly differs for different atmospheric conditions. Results showed that this contribution brings about additional 10 % error on top of the Rayleigh-Brillouin correction. Since the later is already a small contribution to the HLOS error, the impact of the Mie-contamination correction sensitivity is not seen as a significant contribution.

The most appropriate option to quantify the sensitivity to temperature and pressure on HLOS retrievals is to use the best collocated information on temperature and pressure. The study mostly confirms the expected values from (Dabas et al., 2008) who showed the relatively small contribution of the model temperature errors (around 1-2 K for a 24 hour forecast) to the overall HLOS error. However, results have highlighted the necessary quantification of the spatio-temporal variability of these errors that can be locally non negligible. Since the current HLOS random observation errors from the Aeolus mission are significantly

larger than expected due to a number of instrumental drawbacks, there is not much improvement to be expected from a better quantification of model temperature errors in L2Bp. Currently the L2Bp assumes no error in temperature and pressure fields from the IFS model, which, as shown by this study, appears to be a very good approximation for the purpose of NWP data assimilation of HLOS winds. However, it is recognized that uncertainties related to the Rayleigh-Brillouin corrections will become of more significance for the Aeolus follow-on missions where the quality of observations is expected to be improved.

At the same time it is as well recognized that model errors should decrease or be better quantified in the coming years, which again reduces the significance of this particular uncertainty in the HLOS retrieval.



*Data availability.* The reprocessed Aeolus dataset of baseline 2B10 is publicly available at the ESA Aeolus dissemination system. The Aeolus data are publicly available since May 2020.

*Author contributions.* The majority of the set-up of the experiment, the statistical analysis and the publication has been prepared by MŠ.
The framework for the production of the auxiliary meteorological files inside the ARPEGE data assimilation system has been designed by a collaboration of all authors. All co-authors engaged in the discussion, suggestions for improvements and contribution to the writing process of the publication.

*Competing interests.* The authors declare that they have no conflict of interest.

*Disclaimer.* The results presented in section 4.4 are preliminary since data that have been used are not yet publicly available since they have
not been yet properly calibrated and validated.

*Financial support.* This work was supported by grants from the French Space Agency CNES (Centre Nationale des Etudes Spatiales) under contract 5461-MTO-4500064868

*Acknowledgements.* The authors want to thank to the ESA and DISC (Data, Innovation and Science Cluster) for the support and the provision of the preliminary datasets. Especially we would like to thank Michael Rennie (ECMWF) and Jos De Kloe (KNMI) for their support and
feedback during the study.



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
