# Peer review of "Sensitivity of Aeolus HLOS winds to temperature and pressure specification in the L2B processor"

_Atmospheric Measurement Techniques, 2021_

## Author Comment (AC1)

**Response to Referee Comment (RC1) on "Sensitivity of Aeolus HLOS winds to temperature and pressure specification in the L2B processor"**

We are grateful for all the comments and suggestions. We believe that they led to several improvements in the revised text by a better description of the configuration errors and underlying limitations and also by providing a better flow of the information and the study results.

**General Comment:**

*This study examines the sensitivity of retrieved HLOS Rayleigh winds to temperature and pressure fields that should be provided to the Aeolus L2B processor for the Rayleigh-Brillouin correction. The sensitivity to Mie-contamination correction is also assessed. The difference between IFS and ARPEGE short-range forecasts are used to estimate the uncertainties in temperature and pressure fields. These differences are likely to be smaller than real temperature and pressure short-range forecast errors because the IFS and ARPEGE forecast systems are based on the same NWP model and their forecasts rely on the same observing networks. Hence, it is not surprising that the differences found in this study are generally small and lead to L2B HLOS wind variations less than 0.15 m/s in 99% of cases. Although using differences between short-range forecasts from two NWP models as a proxy for the temperature and pressure errors is questionable, there is no obvious and valuable alternative. As such, the article is relevant and represents an interesting follow-on paper of Dabas et al. (2008), which shows that the retrieved HLOS Rayleigh wind is most sensitive to the prescribed temperature and its uncertainty should be less than a few degrees. This requirement is generally met by using short-range forecasts of temperature and pressure from modern NWP models, which is confirmed in the present article.*

*An interesting aspect examined in section 4.3 is the correction of L2B HLOS winds (oper) using differences between the reference temperature from IFS and the short-range forecast temperature from ARPEGE. It is shown that this correction is generally small, but could be significant over some specific locations such as in the stratosphere, near-surface polar regions and jet streams, leading to substantial HLOS wind corrections. However, these corrections remain smaller than the estimated FM-B HLOS wind errors, which is approximately 4 m/s during the period examined. This investigation will be useful to other NWP centers that use the L2B product generated at ECMWF for their research and operational applications.*

*The authors provide many technical details about the processing of Aeolus data at Meteo-France that make the article somewhat difficult to read, especially for readers outside the NWP community. Eliminating some technical details in sections 3.1 and 3.2 would improve the manuscript. A careful revision of the English is also recommended.*

We thank reviewer for this comment. We believe that a clear description of the experiment is crucial for the reproducibility. In this paper we introduced a methodology that is not following the operational production of AUX_MET files since we considered a different data assimilation system. We think that some details of this production must be kept in the paper for any other NWP group that would be interested in running their own L2B processing and AUX_MET production. As well, a clear description is necessary to be able to introduce all error sources affecting the sensitivity study described afterwards. Therefore, we removed some details that are not necessary and rather add some additional explanation on error sources arising from the chosen experimental set-up. English has been revised as well.

**Specific comments:**

**C1:** *Lines 8, 10, 12 and elsewhere in the text: It is stated that model errors for prescribed temperature and pressure are not taken into account in the estimation of HLOS winds. However, what are actually not accounted for are forecast errors, which include both model, representation and initial condition errors propagated in time. It is important to make that distinction because near-surface errors over the poles (model and representation errors) are not of the same nature of errors near jet streams (mostly from initial condition errors). This confusion between model and forecast errors should be corrected throughout the manuscript. Alternatively, this issue could be discussed in the introduction by stating that model error in this article means the sum of model, representation and initial condition errors. Another source of errors could be the difference in the forecast lead-times of IFS and ARPEGE temperature and pressure used in the AUX_MET files (i.e. 6-h and 30-h forecasts for IFS vs 3-h to 9-h forecasts for ARPEGE).*

The confusion on errors affecting this study has been corrected over the whole text. We thank the reviewer for providing this comment since we realized that the a number of factors affecting this sensitivity study were not properly described in the previous version of the text. This has been corrected at several places: lines 41-42 introduce the most well known and expected error sources, lines 56-63 introduce errors related to the comparison of two NWP forecasts and in lines 157-164 additional error sources are described arising due to differences in data assimilation system design and forecast lead time.

**C2:** *Lines 53-55 : The reasons and limitations of using the difference between short-range forecasts from two NWP for assessing the impact of temperature and pressure uncertainties should be further discussed and justified here. In particular, the underestimation of the temperature errors due to error correlation between IFS and ARPEGE short-range forecasts.*

As mentioned for comment C1, a new paragraph has been added to address this issue (lines 56-63). The main reason for using 2 NWP forecasts is the ability to study (fairly economically) the spatial-temporal variability in forecast errors (although due to problems mentioned in lines 56-63 we have been finally forced to study zonally-averaged patterns of uncertainties) which is currently not taken into account in the operational L2B. This study was found important especially for future missions. A better experimental set-up would be to consider the forecast error information from an operational ensemble of forecasts, however, this has not been implemented.

**C3:** *Line 58 : The definition of AUX_MET should be provided here instead of in line 98, or simply replace AUX_MET here by the plain definition 'auxiliary meteorological data input'.*

We agree, it is corrected.

**C4:** *Line 71 : This expression is not a linear interpolation but a first order Taylor expansion. Replacing linear interpolation by linear approximation would be fine.*

We agree, it has been changed.

***C5:*** *Line 129 : AUX_METoper are the met data generated from the IFS short-range forecasts. Hence, it would be easier for the reader to replace AUX_METoper by AUX_METifs or AUX_METecmwf.*

We agree. It has been corrected as well in Fig. 2.

***C6:*** *Line 209 : Is it 89 km or 87 km as usually reported many Aeolus studies?*

This has been corrected to the value that is reported by *Martin et al., Validation of Aeolus winds using radiosonde observations and numerical weather prediction model equivalents, Atmos. Meas. Tech., 14, 2167–2183, https://doi.org/10.5194/amt-14-2167-2021, 2021.*

***C7:*** *Lines 212-214 : Do you also consider a background check quality control?*

We thank the reviewer for making a comment on this, we forgot to add it in the first version of the text. It has been corrected.

***C8:*** *Lines 262-264 : It should be mentioned that the temperature difference between IFS and ARPEGE are expected to be smaller than the real temperature uncertainty because of the similarities between the two forecast systems.*

We agree. This has been modified in the revised text (lines 56-63).

***C9:*** *Line 284 : I suggest inverting Fig. 7 and Fig.6 to make these figures appearing in the right order in the text.*

We agree with this suggestion that improves the readability.

***C10:*** *Lines 289-290 : It is hard to say that the granulation is due to model error because the truth is unknown. What we see here is the difference between two background fields.*

We agree this has been modified.

***C11:*** *Line 358 : 'Regarding the results presented further this has been found insignificant'. This sentence is not clear.*

We agree that this sentence has been poorly written. The idea is that the difference is so small that it is not relevant for the remaining of the study. This is corrected in revised text.

***C12:*** *Line 399 : Could also be due to differences between the IFS and ARPEGE data assimilation systems.*

We agree with this possibility. We changed the text accordingly.

**C13:** *Lines 400-401 : This comparison between the short-range forecast differences and the EDA spread should be discussed earlier in the text (e.g. lines 262-264). This is an important element to justify the use of temperature differences between the IFS and ARPEGE forecasts as a proxy for reference temperature errors.*

We agree with the suggestion. The corresponding discussion has rather been moved in section 4.2 (lines 304-309), that introduces temperature and pressure differences.

---

## Author Comment (AC2)

**Response to Referee Comment (RC2) on "Sensitivity of Aeolus HLOS winds to temperature and pressure specification in the L2B processor"**

We are grateful for the comments and suggestions.

**General Comment:**

*This manuscript presents a study on the sensitivity of Aeolus HLOS wind retrieval to temperature and pressure in NWP models used in the L2B processor. This is an interesting study because it is important to have a good characterization of uncertainties in observations to assimilate them in NWP systems. In order to estimate correctly the HLOS sensitivity in the Rayleigh-Brillouin channel, it is necessary to know the temperature and the pressure. These quantities are estimated using the information provided by NWP models. The study confirms that in more than 99% of the cases, the impact of temperature and pressure errors have a negligeable impact on HLOS wind retrieval taking into account the relatively large errors of Aeolus HLOS data. However, it will be necessary to better estimate this impact for Aeolus follow-on mission where the expected quality of the observations will be hopefully improved. The originality of the approach is to estimate the errors in NWP temperature and pressure fields from the difference between two NWP models IFS and ARPEGE. However it is not obvious that the difference between two NWP models is really representative of the model errors. This assumption needs to be discussed in the manuscript.*

*I agree to anonymous reviewer #1 to consider that some technical details in section 3.1 and 3.2 could be removed to render the paper easier to read for non-specialists of NWP data assimilation.*

*Despite these remarks, I consider that the paper brings new and useful information on characterization of Aeolus HLOS wind retrieval.*

We thank reviewer for this comment. We believe that a clear description of the experiment is crucial for the reproducibility. In this paper we introduced a methodology that is not following the operational production of AUX_MET files since we considered a different data assimilation system. We think that some details of this production must be kept in the paper for any other NWP group that would be interested in running their own L2B processing and AUX_MET production. As well, a clear description is necessary to be able to introduce all error sources affecting the sensitivity study described afterwards. Therefore, we removed some unnecessary details and rather add some additional explanation on error sources arising from the chosen experimental set-up.

We are aware of limitations regarding the estimation of uncertainties from two NWP forecasts. This has been discussed in a greater detail in lines 56-63 of the Introduction section in the revised version. The main reason for using 2 NWP forecasts is the ability to study (fairly economically) the spatial-temporal variability in forecast errors (although due to problems mentioned in lines 56-63 we have been finally forced to study zonally-averaged patterns of uncertainties) which is currently not taken into account in the operational L2B processor. This study was found important especially for future missions for which a statement is given in the conclusion of our study. A better experimental set-up would be to consider the forecast error information from an operational ensemble of forecasts, however, this has not been implemented.

**Specific comments:**

***C1:*** *Line 36: I do not understand the comment on the deviation of Rayleigh-Brillouin deviation from the Gaussian spectrum. Is it not due to acoustic waves rather than atmospheric stratification?*

We agree that this sentence was poorly written. This deviation is due to the increased collision between molecules and the induced acoustic waves. It is corrected in the revised text.

***C2:*** *Lines 217: Please explain what is the median absolute difference (difference between percentiles 75 and 25). This quantity is not so frequently used in the atmospheric community.*

Clarification is added on the use of mad instead of std.

---

## Author Response (AR2)

**Response to Associate Editor Comment on "Sensitivity of Aeolus HLOS winds to temperature and pressure specification in the L2B processor"**

We are grateful for the additional comment from the Associate Editior.

**Comment:**

*This paper provides a valuable assessment of the sensitivity of the Aeolus Level 2 horizontally projected line-of-sight (HLOS) wind accuracy to a priori temperature and pressure information. The authors have responded very well to the comments and requested updates by the reviewers. This paper is therefore recommended for publication subject to minor revisions, based on the following editorial comment:*

*In line 307-308, I propose to replace "… are rather similar to those represented by the EDA (in particular Fig. 5a of Isaksen et al. (2010)), comforting our assumptions." with "… are relatively similar to those represented by the EDA temperature spread in Fig. 5a of Isaksen et al. (2010), supporting the experimental approach use in this paper. A qualitative comparison of the figures also show that the magnitude of the EDA temperature spread in Fig. 5a of Isaksen et al. (2010) is typically XX larger than the differences reported in Figure 5, with peak values of XX in xxx regions." The second sentence can be reformulated as needed, and the authors are asked to fill in the XX and xxx according to their qualitative analysis. A further sentence should also be added commenting how this compares to the temperature errors used for the further analysis in this paper. The discussion in the paragraph from line 409-420 should also be updated accordingly.*

We thank the Associated Editor for this suggestion. This brings an additional level of information that better described the expected underestimations of the errors described in the paper. The suggested corrections have been implemented in the modified version of text (lines 304-311 and lines 419-420).